# Using Macro- and Microscale Preservation in Vertebrate Fossils as Predictors for Molecular Preservation in Fluvial Environments

**DOI:** 10.3390/biology11091304

**Published:** 2022-09-02

**Authors:** Caitlin Colleary, Shane O’Reilly, Andrei Dolocan, Jason G. Toyoda, Rosalie K. Chu, Malak M. Tfaily, Michael F. Hochella, Sterling J. Nesbitt

**Affiliations:** 1Department of Geosciences, Virginia Tech, Blacksburg, VA 24061, USA; 2Cleveland Museum of Natural History, Cleveland, OH 44106, USA; 3Atlantic Technological University, ATU Sligo, Ash Lane, F91 YW50 Sligo, Ireland; 4Texas Materials Institute, University of Texas at Austin, Austin, TX 78712, USA; 5Environmental Molecular Sciences Laboratory, Pacific Northwest National Laboratory, Richland, WA 99354, USA; 6Department of Environmental Science, University of Arizona, Tucson, AZ 87519, USA; 7Earth Systems Science Division, Pacific Northwest National Laboratory, Richland, WA 99354, USA

**Keywords:** molecular taphonomy, fossils, preservation, mass spectrometry, dinosaurs

## Abstract

**Simple Summary:**

Fossils are the only direct evidence of life throughout Earth’s history. We examine the biology of ancient animals to learn about evolution and past ecosystems. Biomolecules are a relatively new source of information from fossil records because new technology is now being used in paleontology that makes it possible to detect molecular remains in fossils. However, molecules extracted from fossils are complex mixtures with environmental and other sources of organic compounds. Additionally, macroscale preservation is well-known to vary greatly across fossil localities. Therefore, a goal in molecular paleontology is to develop ways to predict where molecules may be preserved and differentiate between endogenous and exogenous sources. Here, we use a powerful combination of methods that focus on high-resolution mass spectrometry to evaluate the molecular-scale preservation of a dinosaur quarry from the Triassic Period. We found that despite very good overall preservation at this locality, there is no evidence of endogenous molecules, demonstrating that molecular preservation is variable and that good macro- and microscale preservation cannot necessarily be used as predictors for biomolecule preservation in the fossil record.

**Abstract:**

Exceptionally preserved fossils retain soft tissues and often the biomolecules that were present in an animal during its life. The majority of terrestrial vertebrate fossils are not traditionally considered exceptionally preserved, with fossils falling on a spectrum ranging from very well-preserved to poorly preserved when considering completeness, morphology and the presence of microstructures. Within this variability of anatomical preservation, high-quality macro-scale preservation (e.g., articulated skeletons) may not be reflected in molecular-scale preservation (i.e., biomolecules). Excavation of the Hayden Quarry (HQ; Chinle Formation, Ghost Ranch, NM, USA) has resulted in the recovery of thousands of fossilized vertebrate specimens. This has contributed greatly to our knowledge of early dinosaur evolution and paleoenvironmental conditions during the Late Triassic Period (~212 Ma). The number of specimens, completeness of skeletons and fidelity of osteohistological microstructures preserved in the bone all demonstrate the remarkable quality of the fossils preserved at this locality. Because the Hayden Quarry is an excellent example of good preservation in a fluvial environment, we have tested different fossil types (i.e., bone, tooth, coprolite) to examine the molecular preservation and overall taphonomy of the HQ to determine how different scales of preservation vary within a single locality. We used multiple high-resolution mass spectrometry techniques (TOF-SIMS, GC-MS, FT-ICR MS) to compare the fossils to unaltered bone from extant vertebrates, experimentally matured bone, and younger dinosaurian skeletal material from other fluvial environments. FT-ICR MS provides detailed molecular information about complex mixtures, and TOF-SIMS has high elemental spatial sensitivity. Using these techniques, we did not find convincing evidence of a molecular signal that can be confidently interpreted as endogenous, indicating that very good macro- and microscale preservation are not necessarily good predictors of molecular preservation.

## 1. Introduction

If original biological compounds (biomolecules) are preserved on long timescales (>10 million years), then more of the biological remains of extinct organisms can be uncovered from the fossil record than previously considered possible, expanding what is known about ancient biology and taphonomic processes. Studies of the preservation of biomolecules often focus on exceptionally preserved fossils that retain soft tissues ([1] and references therein); however, exceptional preservation (e.g., hair, feathers, skin) requires specific conditions to occur (including sediment chemistry and microbial activity) and is rare [2]. Therefore, most fossils, particularly those of terrestrial vertebrates which are often entirely skeletal remains, are not often considered exceptionally preserved. The quality of preservation of bone varies from intact well-preserved, articulated skeletons to weathered bone fragments [3,4,5]. The characterization of good preservation is dependent on scale: (1) good macro-level preservation is the presence of articulated skeletons or features on the bones (e.g., muscle scars); (2) good micro-level preservation is the retention of the microstructures in bone that are often examined in histological studies (e.g., external fundamental systems, three-dimensionally preserved canaliculi); and (3) good molecular-level preservation is the retention of original biomolecules (e.g., nucleic acids, proteins, lipids). With the increased use of high-resolution mass spectrometry, more studies on terrestrial vertebrate fossils have begun to demonstrate that biomolecules preserve more readily than previously considered and in a greater range of depositional settings [5,6,7,8,9,10], opening much of the vertebrate terrestrial record to recovering more data for ancient animals [11], even from weathered bone fragments, which have traditionally been considered poorly preserved [12].

The terrestrial fossil record is heavily biased toward fossils preserved in fluvial sedimentary settings and the majority of studies that have investigated the preservation of biomolecules in dinosaurs and other terrestrial vertebrates have done so in fossils that are preserved in stream channel, flood plain, delta, and coastal paleoenvironments [5,6,13,14,15]. Course-grained sandstones and conglomerates are common lithologies in these paleoenvironments and are not normally considered to be conducive to exceptional or good preservation [16], although some have suggested that the porosity of sandstones may improve molecular preservation [6]. Additionally, the influence of water on molecular-scale preservation in fluvial environments has not been experimentally tested, despite being hypothesized to not be conducive to the preservation of biomolecules [17] and recent studies have reported molecular preservation in marine depositional environments [7,8].

Therefore, to explore the molecular preservation of a fluvial terrestrial fossil assemblage, we examined the Hayden Quarry (HQ) (Chinle Formation, Petrified Forest Member, Ghost Ranch, NM, USA), a locality that preserves an extraordinary Late Triassic (~212 Ma) record of dinosaur evolution and paleoenvironmental change [18,19,20]. The Petrified Forest Member is a series of paleo-channels, with alternating mudstones and siltstones and poorly sorted sandstones and conglomerates [19] and the HQ is divided into four active quarries (H1–H4). The depositional environment is interpreted as episodes of transient flooding, along with periods of standing water and preserves terrestrial and semiaquatic animals [18], (Appendix A). The HQ has a diverse assemblage of terrestrial vertebrates, with high-fidelity preservation at both the macro-scale (i.e., complete skeletons and very small vertebrates, with vertebrae discovered as small as ~1 mm) and microscale, with histological analyses showing high-fidelity preservation of bone microstructures [21].

Incorporating taphonomy into fossil studies is essential for understanding the geological history of the samples to make predictions about what may be preserved [22]. Here, we examine whether high-quality macro- and micro-level preservation are good indicators of high-quality molecular-level preservation. To examine the organic preservation of fossils preserved in a fluvial environment, we used mass spectrometry to examine animal biomolecules (lipids and amino acids) and environmental compounds (e.g., phenol). We analyzed three different types of fossilized tissues from one paleo-channel in the HQ (H4): the femur of the early theropod dinosaur *Tawa hallae* (GR1065) [23], a phytosaur tooth (GR1064), and a coprolite with digested bone from an indeterminate vertebrate (GR1063). We also analyzed an additional bone from H4 (GR1066) to test for any differences between analyzing thin section and non-thin sectioned bone using these techniques. Additionally, we compared the Triassic fossil bone to dinosaur bones from the Cretaceous and Jurassic Periods from similar depositional environments, bone from extant vertebrates, experimentally matured bone (i.e., bone from extant vertebrates that was subjected to a range of temperatures to accelerate the degradation process), and a matrix sample as a control.

Institutional Abbreviations: Ghost Ranch, Hayden Quarry (GR), Los Angeles Museum of Natural History (LACM), Mammoth Site of Hot Springs, South Dakota (MS).

## 2. Materials and Methods

### 2.1. Specimens

Thin sections from the Hayden Quarry (HQ) were analyzed to compare different types of fossils and to test the differences in analyzing thin sections and whole bone fragments using surface mass spectrometric techniques (e.g., time of flight secondary ion mass spectrometry). The fossils were embedded in CastoliteAP, a clear polyester resin (vacuumed to remove bubbles). After curing, a Buehler (IsoMet 4000 Lake Bluff, IL, USA) saw with a diamond wafering blade was used to create thin sections that were subsequently glued to glass slides with Aron Alpha (Type 201) cyanoacrylate. The excess material was ground down using a Hillquist thin-section machine and hand polishing on an Ecomet 2 speed grinder-polisher to a thickness that the microanatomy could be viewed using a light microscope. The HQ specimens thin sectioned and analyzed include: a tooth (phytosaur; GR1064), a bone (femur from the early theropod *Tawa hallae*; GR1065) and a coprolite from an unidentified vertebrate (GR1063) collected in Hayden Quarry 2 (H2). Additionally, dinosaur rib fragments used in analysis include a theropod from the Hell Creek Formation in Montana (Hell Creek), USA (LACM 23844, Late Cretaceous Period), a sauropod from the Morrison Formation in Utah (Morrison), USA (LACM 154089, Late Jurassic Period), and two dinosaur rib fragments from Hayden Quarry 4 (H4) (GR1065, GR1066, Triassic Period). All of these fossils were weathered out of or excavated from fluvial depositional environments with sandstone or mudstone lithologies [13,18,24]. Specimens were excavated in situ and areas sampled showed no signs of surface weathering. A matrix sample collected in conjunction with the rib fragment (GR1066) from H4 was included to compare to the compounds found in the fossils. Recently deceased alligator (*Alligator mississippiensis*, TMM M-12613) and elephant (*Loxodonta*, MS-E01) rib bone samples were used to compare unaltered bone chemistry using the same techniques (Table 1).

### 2.2. Maturation Experiments

Experimentally matured bone samples (modern elephant, MS-E01) were reanalyzed from a previous study [25]. A diamond saw (Dremel**^®^**) was sterilized to prevent contamination and used to cut a fresh elephant rib bone (deceased zoo animal, ME-E-01) into three 2 mm^2^ fragments. The fragments were sealed in 3 mm × 15 mm platinum capsules; however, this does not prevent water from evaporating. They were loaded into cold sealed pressure vessels in the Hydrothermal Laboratory at Virginia Tech. These short-term experiments accelerate the degradation of the bone and were conducted for 24 h at 100 °C, 200 °C and 250 °C at atmospheric pressure (based on the protocol in [26]).

### 2.3. Time-of-Flight Secondary Ion Mass Spectrometry (TOF-SIMS)

TOF-SIMS analysis was performed using an ION-TOF GmbH, Germany TOF.SIMS 5 at The University of Texas at Austin, Texas Materials Institute. A pulsed (20 ns, 10 kHz) analysis ion beam of Bi_3_^+^ clusters at 30-kV ion energy was raster-scanned over 500 × 500 µm^2^ areas. Bi_3_^+^ polyatomic sputtering was used to reduce the fragmentation of large organic molecules. A constant flux, 21 eV electron beam was used during data acquisition to reduce sample charging. Secondary ions had positive polarity and an average mass resolution of 1–3000 (m/δm). The base pressure during acquisition was <1 × 10^−8^ mbar. Mass calibration was performed by identifying the peak positions of CH_2_^+^, CH_3_^+^, C_2_H_3_^+^, and C_3_H_3_^+^ secondary ions. Regions of interest were chosen to reduce the effects of topography (which even at micron-scales, can influence the time it takes certain molecules to reach the analyzer, which degrades the mass resolution, thereby impeding correct molecular assignment).

The benefit of surface mass spectrometry is it is minimally destructive and can be used to examine the spatial distribution of fossils. One of the drawbacks of using this technique when evaluating protein preservation is that it does not provide certain types of data (e.g., peptide sequences). Therefore, we determined a chemical fingerprint of 86 peaks for protein degradation products (amino acids and amino acid fragments) and the inorganic components of bone. We mapped the distribution of positive spectra of ionized molecules on the bone surface using TOF-SIMS in fresh bone and dinosaur fossils (Table 1, Figure 1). Amino acids, amino acid fragments and mineral elements were chosen to characterize the preservation of each sample. The fingerprint was developed by combining relevant peaks from previous analyses [25,27] and choosing additional peaks that are present in the samples (Appendix A). Matrix (rock samples not containing fossils) from the same horizon as the fossils in the HQ and fresh bone were used to compare the degradation of bone that occurs during fossilization.

### 2.4. Lipid Analyses

Two Triassic Period rib fossils (GR1066, GR1067) were separated from matrix sediment manually and the fossil surfaces were cleaned using a dental drill and solvent-cleaned steel drill- bits. Powders were drilled from cleaned fossils. The powder from the cleaning procedure for H2 was also retained. Sediment matrix was powdered using a solvent-cleaned mortar and pestle. Between 250 and 600 mg of powdered sample was each weighed into 12 mL glass tubes. Samples were extracted for 30 min (ratio of solvent: sample was 5:1) with 9:1 (*v/v*) dichloromethane/methanol [28] using sonication in an ultrasonic bath at room temperature (~21 °C). The extract was separated from solid residue by centrifugation. Extracted residues were re-extracted with fresh solvent for a total of three extraction and supernatants from each step were combined to give a total lipid extract (TLE). TLEs were concentrated to minimal volume under a gentle stream of high-purity N_2_ gas. A portion each TLE was silylated with N,O- Bis(trimethylsilyl)trifluoroacetamide /trimethylchlorosilane mixed with pyridine (9:1 *v/v*) at 70 °C for 2 h. A portion each TLE was reacted with N,O-Bis(trimethylsilyl)trifluoroacetamide/trimethylchlorosilane mixed with pyridine (9:1 *v/v*) at 70 °C for 2 h. This reaction replaces active hydrogens on polar functional groups (e.g., hydroxy groups) with a trimethylsilyl moiety to increase analyte volatility and thermal stability, thereby ensuring functionalized lipids are amenable to GC analysis. Aliquots of the derivatized samples were analyzed by gas chromatography/mass spectrometry (Agilent 5890 GC hyphenated to an Agilent 5975C Mass Selective Detector). The GC was equipped an Agilent J&W HP-5MS non-polar capillary column (30 m length, 0.25 mm inner diameter, 250 µm film thickness). The GC temperature program was: 70 °C for 2 min, ramp at 10 °C min^−1^ to 130 °C, followed by a ramp to 300 °C at 4 °C min^1^ and a final hold time of 20 min. The mass spectrometer was operated in electron impact ionization mode (70 eV), with a mass scan range from *m*/*z* 50 to 600. All glassware was fired (550 °C overnight) and all solvents used were high-purity (OmniSolv). Procedural blanks were run to monitor background contamination.

### 2.5. Fourier-Transform Ion Cyclotron Resonance Mass Spectrometry (FT-ICR MS)

Mortar and pestles were washed, then wrapped in aluminum foil, rinsed in nano pure water, rinsed in ethanol and then were combusted at 400 °C for 8 h. The aluminum foil remained on the pestle and lined the mortar throughout grinding to prevent contamination. Once they cooled down, they were used to powder the fossil samples. Blanks were created by rinsing and collecting 2 mL of nano pure water, 2 mL of MeOH and ~2 mL of CHCl_3_ from each mortar and pestle. We analyzed bone and sediment from HQ, two dinosaur bones (LACM 154089 and LACM 154089) and a modern elephant bone (MS-E-01). We combined two rib fragments from Ghost Ranch (GR1066, GR1067) as one sample. Each sample was powdered using the mortar and pestle until we had ~1.5 g for each sample. The elephant sample was prepared in liquid nitrogen prior to powdering (Table 1).

Three sets of extractions were conducted on each bone sample: water, methanol (MeOH,) and chloroform (CHCl_3_) because each solvent is elective for a specific type of organic compound based on its polarity [29]. Water extractions were done first to extract water-soluble small molecules such as sugars, amino sugars and amino acids. 5 mL of nano pure water was added to the powdered samples in leach-free falcon tubes and vortexed at 1000 rpm for 2 h. Samples were then centrifuged for 5 min at 4500 rpm to pellet the samples. These steps were repeated using methanol (MeOH) to sequentially extract other semi polar organics such as lignin-like compounds with both polar and non-polar sides due to its hyperbranched structure and chloroform (CHCl3) to extract non-polar lipids [30]. The samples were then stored overnight at 4 °C. Prior to infusion into the mass spectrometer the water extracted samples were acidified to a pH of 2, concentrated and desalted using Bond Elut PPL cartridges and following procedures from [31]. The methanol extracted samples were run without clean up and the chloroform samples had methanol added in a 1:1 (v:y) to aide in ionization.

The mass spectrometry analysis was performed using a 12T Fourier transform ion cyclotron resonance mass spectrometer (FT-ICR MS) (Bruker solariX, Billerica, MA, USA) outfitted with a standard electrospray ionization (ESI) interface. Samples were directly infused into the mass spectrometer using a 250 μL Hamilton syringe at a flow rate of 3 μL/min. The coated glass capillary temperature was set to 180 °C and data were acquired in positive and negative mode for better overall coverage of detected molecules. The needle voltage was set to +4.2 kV negative mode and −4.4 kV in positive mode. The data were collected by co-adding 200 scans with a mass range of 100–900 *m/z*, at 4 M with a resolution of 240 K at 400 *m/z*. Formulae were assigned by first converting the raw spectra into a list of peaks using Bruker Data Analysis (version 4.2) and applying an FTMS peak picker with a signal-to-noise ratio set to 7 and absolute intensity set to 100. Once the raw spectra were converted to a list of peaks and their resulting mass-to-charge (*m/z*) ratio data was internally calibrated, formula assigned and peaks aligned using Formularity [32] and following the Compound Identification Algorithm [33,34]. Predicted chemical formulae were assigned with C, H, O, N and S and excluding all other elements with the following rules: O > 0 AND N <= 4 AND S < 2 AND 3 * P <= 0. Alignment tolerance was set to 0.5 ppm and calibration tolerance was set to 0.1 ppm. The molecular formulae in each sample were evaluated on van Krevelen diagrams [35], based on their H:C ratios (*y-*axis) and O:C ratios (x-axis) assigning them to the major biochemical classes (e.g., lipid, protein, lignin, carbohydrate, etc.) according to [36]. The H:C and O:C ranges for biochemical classification are provided in Appendix A.

## 3. Results

In this study, we analyzed different fossil types from a single fossil locality (HQ) using various analytical methods for evaluating the preservation of different molecular classes, as well as inorganic signals to understand the overall molecular taphonomy of the Hayden Quarry. To test if different types of fossils (i.e., bone, tooth, coprolite) can be chemically distinguished from one another, we used surface mass spectrometry (TOF-SIMS) and compared the samples using multivariate statistics. The principal component analysis (PCA) compared the three different fossil types (i.e., bone, tooth, coprolite), the additional dinosaurian fossils, the unaltered bones of extant vertebrates and the experimentally matured bone to determine the variance between each of the samples (Figure 1). Additionally, we included both thin sections and one whole bone sample from the HQ to see if embedding them in resin altered the molecular signal. All the HQ samples including both thin sections and the bone sample plot together, demonstrating no change in the molecular signal from embedding them. When comparing the samples, there is little variation between all of the fossils from the Hayden Quarry (thin sections and polished bone fragment), which all plot together, along with the 250 °C matured bone. Although, the 200 °C matured bone and the Morrison bone do show slight variation from the others and plot more closely to the sediment matrix sample from the HQ. The HQ matrix sample does show a greater amount of variance from the HQ fossil samples. The two unaltered bones from extant vertebrates (elephant and alligator) and the 100 °C matured bone have similar molecular signatures to one another but differ from the rest of the samples. TOF-SIMS molecular maps (Figure 2) of the HQ polished bone fragment (GR1066) show that the presence of specific elements and molecules varies between certain bone features. Calcium (Ca) and iron (Fe) are ubiquitous across the bone surface but show a decreased abundance in certain areas (around microstructures in the bone) that have a higher abundance of strontium (Sr). Bone microstructures do have macroscopic evidence of mineral infilling. There is one area of the bone that shows a low abundance of Ca, Fe and Sr and a high abundance of the amino acids glycine (Gly) and alanine (Ala). Otherwise, these amino acids are in extremely low abundance, if present at all, across the bone surface.

To evaluate the presence of additional biomolecules, we examined lipids in fossil bone and the surrounding matrix (Figure 3). Lipids were restricted to fatty acids, including unsaturated fatty acids and the lipid profiles were very similar between the fossils and the matrix. The associated matrix had more than eight times the extractable lipids than the fossil bone. Additionally, we compared the fossil, bone and matrix samples by evaluating the presence of additional organic molecules present in the HQ using high-resolution FT-ICR MS. In the PCA analysis (Figure 4), the placement of the elephant and mammoth bones were most heavily influenced by lipid- and protein-like compounds, the Hell Creek and Morrison bones were most heavily influenced by oxygenated, phenol- and amino sugar-like compounds and both the HQ bone and HQ matrix showed little variation from one another and placement was most heavily influenced by unsaturated- and condensed hydrocarbon-like compounds. FT-ICR MS at high magnetic fields provides detailed molecular information about complex mixtures, like those found in fossils, due to its high resolution and mass accuracy, a requirement to assign unique, unambiguous molecular formulae to each peak across an entire molecular weight distribution (200 < *m/z* < 1500) Therefore, the absence of endogenous material in the samples tested in this study suggests that these compounds are indeed not present.

## 4. Discussion

Depositional environment plays a large role in fossil preservation [25,37,38]. Fluvial formations are common sources of fossil material, particularly of terrestrial vertebrates. The variation in preservation in fluvial burial environments is indicative of the variation in the environments themselves and leads to extremes in preservation, ranging from complete skeletons to fragmentary bone [39]. Therefore, molecular preservation in fluvial environments likely varies as much as macro-scale vertebrate preservation. Amino acids and amino acid fragments were detected in dinosaurian fossils dating back to the Late Triassic (~212 Ma) and the presence of amino acids, specifically glycine, has been included as supporting evidence of ancient collagens [5,6,7]. We detected an amino acid signature unique to the fossil bones, teeth and coprolites and distinct from the surrounding matrix. However, the absence of lipids and the main signal of condensed hydrocarbon in the HQ bone and matrix, which are both very similar, cast doubt that the amino acids detected in the HQ fossils have an animal origin.

Molecular analytical techniques being applied to fossil studies often fall into two general categories: (1) those that extract targeted compounds and inject them into a mass spectrometer (e.g., DI, LC-MS) to examine what is preserved in the fossil (e.g., paleoproteomics) [40] and (2) those that use surface analytical techniques (e.g., Raman, TOF-SIMS) to analyze the entire fossil and examine taphonomy (e.g., degradation products, mechanisms of preservation) [41]. To date, no single technique excels at accomplishing both of these goals. The benefit of using surface-sampling techniques includes the ability to examine the spatial distribution of molecules that may vary in different parts of a fossil; however the trade-off is a lack of resolution regarding the compounds themselves. TOF-SIMS specifically, while of great utility in semi-non-destructive fossil analyses on heme and pigments, is not the best mass spectrometric technique for evaluating proteins in fossils ([42] and citations therein). However, the ability to evaluate the structure of the bone and make comparisons between the fossil and the matrix are strong additions to molecular taphonomy studies interested in exploring where molecular data may be best preserved for targeting future analyses and for examining the distinction or interplay between fossil and matrix. TOF-SIMS is a highly element sensitive and selective technique, both chemically and spatially; however, in this application, the amino acids we were examining are relatively small molecules with limited fragmentation patterns, which likely produce secondary ion fragments that generally match the fragments generated by many other organic materials in the environment. This highlights a continuing challenge in paleomolecular studies, which is determining the source of organic material in extremely organic-rich, complex environments.

The methods used in this study were chosen to compare the overall molecular taphonomy of the HQ. When examining the variation between the specimens, we used TOF-SIMS to develop a chemical fingerprint of 86 amino acid and amino acid fragments and found this was a good way of distinguishing between fossil and extant bone and between the fossils and associated matrix, but not between the different types of fossils. This demonstrates that there is a similar organic signal in all of the HQ fossil material that is distinct from the associated matrix and is also distinct from the dinosaur bones from the Cretaceous and Jurassic sites. Therefore, because of the difference between the fossils, the molecular signal is not indicative of generalized contamination. However, because there is no distinction between the bone, tooth and coprolite in the HQ, this signal is likely not evidence of ancient animal remains but may instead be taphonomic or environmental. Recent work has suggested that fossil bone may be a good host to modern microbial growth [43] but we did not find evidence of microbial lipids that would suggest that is the case here. The ability to confidently assign the remnants of biomolecules to ancient animals, ancient microbes or modern microbes will remain one of the major goals of ancient molecular studies, particularly on such long timescales. The experimentally matured bone (all MS-E01, heated to increasing temperatures) depict an interesting alteration in the chemical signature. When compared to the fossils, the molecular signal in the heated bones is altered in a predictable way, with the increase in temperature correlating to the age of the fossils. Heating the bone to 100 °C caused some alteration, but it is still most comparable to the modern bone. Heating the bone to 250 °C led to a molecular signal similar to that found in the HQ bone, which we are interpreting as the absence of a molecular signal from the bone. It is possible that the alteration seen in the modern bone is related to the evaporation of water during the heating process and is worthy of further exploration.

Fossils and sedimentary rocks of this age would typically contain the end products of lipid diagenesis—hydrocarbon skeletons such as steranes or *n*-alkanes—as major extractable compounds [44] and are much more likely to be from modern (or very young) sources. Previous work has shown that lipids can be transported from fossils to surrounding matrices [45] or the opposite, from matrices to fossil bone [43]. Thus, comparison between fossils and matrices must include assessment of the relative abundances, occurrences (presence or absence) and distributions of individual lipids (or classes of lipids). In contrast to Liebenau et al. (2015) results, extractable lipids in our specimens were much higher in the matrix sample than fossils. Given that the distributions of lipids in fossil specimens were very similar to the matrices and the observed concentration gradient, it is more likely that detected lipids in the fossils are exogenous. Given, the presence of exceptionally labile lipids (monounsaturated fatty acids) as major lipids in the fossils analyzed, the similar lipid profiles between fossils and matrices and the much higher concentration of lipids in the matrices, it is likely that the lipids detected in the fossil bones were sourced from the matrix and transported in and that they are from modern/recent biological sources. It is possible that the fatty acids detected are from microbial communities that have colonized the fossil bone matrices, as proposed by Saitta et al. (2018). Lipid data casts additional doubt on the source of the amino acid signature being detected.

We also examined the inorganic components of the bone and additional sources of organics, including environmental contaminants (Figure 4). High levels of phenols were found in the Morrison and Hell Creek fossils, which could be a signal of lignin, representing contamination from vascular plants. Lignin is a biopolymer that preserves on very long timescales (~315 Ma) [17]; however, it is not possible to determine with these methods if it is ancient or recent, although a more recent source of the lignin may be from plant roots seeking out sources of phosphorous from buried bone. The source of phenols may also be microbial because microbes produce phenols when under stressed conditions [46]. The HQ samples had hydrocarbons present in high abundance and the bone and matrix samples were similar in composition. Additionally, iron is present in high intensity across the surface of the HQ bone fragment, while other elements like strontium are isolated to pore spaces in the bone (Figure 2). The high intensity of iron is consistent with proposed conditions in burial environments that may favor soft tissue preservation ([12] and references therin) and strontium levels have been shown to increase with time and during diagenesis [47]. The molecular composition of all of the fossils is similar despite being from different sources (e.g., bone, tooth, coprolite), time periods and burial environments, which may be evidence for a taphonomic source of organics being introduced to fossils that are found in similar depositional environments. Additional analyses that target specific compounds may be able to determine the source, specifically extraction-based proteomic techniques.

## 5. Conclusions

Despite very good macro and micro-level preservation at the Triassic Hayden Quarry, there is no direct evidence for original biomolecules preserved in the fossils. Amino acid evidence shows variation between the fossils and the matrix. This alone is not definitive evidence of original biomolecular preservation; however, we were unable to determine the source of these amino acids with the methods used in this study. Lipid data have no indication of animal lipids and show no distinction in lipid content between the fossil bone and matrix, and the FT-IRC MS data show that the HQ bone and matrix are similar to one another and high in hydrocarbons. Therefore, we have concluded that we detected no original organic preservation at this site. The Hayden Quarry is an example of a fluvial burial environment where we find remarkable macro- and microscale preservation, but no evidence of molecular-scale preservation using these methods. Therefore, as molecular fossil studies continue with the goal of finding trends in preservation to target specimens with probable molecular-scale preservation, we have to continue to consider that variability across fossil localities, which is well-documented at the macro- and microscales, makes it very difficult to make overarching rules about molecular preservation based on burial environment.

## Figures and Tables

**Figure 1 biology-11-01304-f001:**
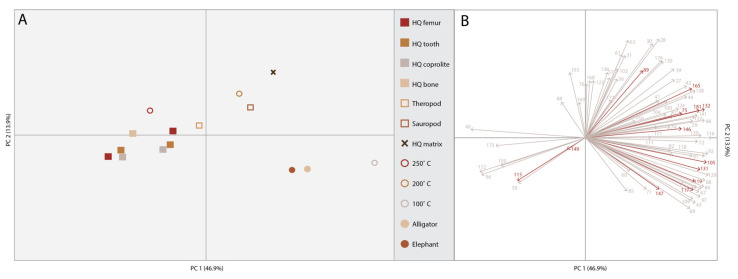
Principal component analysis. (**A**) PCA of chemical signature of 86 organic and inorganic peaks. HQ femur, HQ tooth and HQ coprolite are all thin sections from H2, and HQ bone is an untreated bone fragment from H4. The HQ matrix is from H4. The modern bones (alligator and elephant) and the 100 °C are more similar chemically to one another than the other samples. The HQ fossils show little variation between one another, but there is a greater amount of variation with the associated matrix. Therefore, there are amino acids present in the fossils that are not present in the associated matrix. (**B**) PCA Loadings show how each one of the 86 peaks influences the specimen placement in the PCA.

**Figure 2 biology-11-01304-f002:**
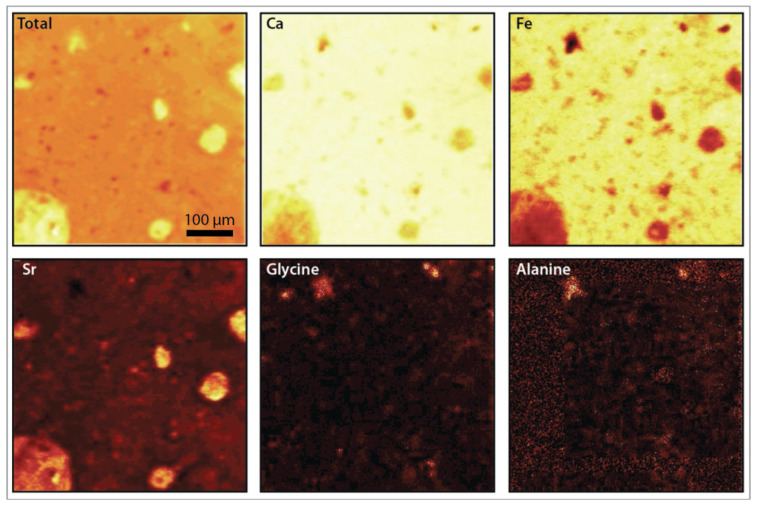
TOF-SIMS images of HQ bone fragment (GR1066). Each image is the same 500 µm × 500 µm area of the bone. The lighter areas represent a greater concentration of a given element or molecule, while the darker areas represent a lower concentration. The total map shows all 86 peaks in the analysis, showing a higher concentration in pore spaces. The rest of the maps show the distribution of specific elements and molecules. Specific features of the bone, like pore spaces, have a higher concentration of strontium (Sr) and lower concentrations of iron (Fe) while some elements, like calcium (Ca) are ubiquitous across the sample surface. Amino acids like glycine and alanine are both present in a single area of the bone.

**Figure 3 biology-11-01304-f003:**
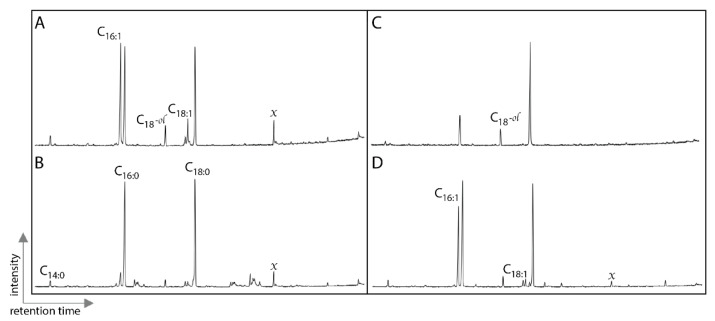
Partial total ion chromatograms of silyated total lipid extracts from Ghost Ranch Hayden Quarry 4: (**A**) fossil bone and (**B**) associated matrix. Hayden Quarry 2 (**C**) fossil bone and (**D** associated matrix. Tetradecanoic acid (C14:0), hexadecanoic acid (C16:0, C16:1), octadecanoic acid (C18:0, C18:1, octadecan-1-ol (C18:0*^-ol^*) and contaminants (x).

**Figure 4 biology-11-01304-f004:**
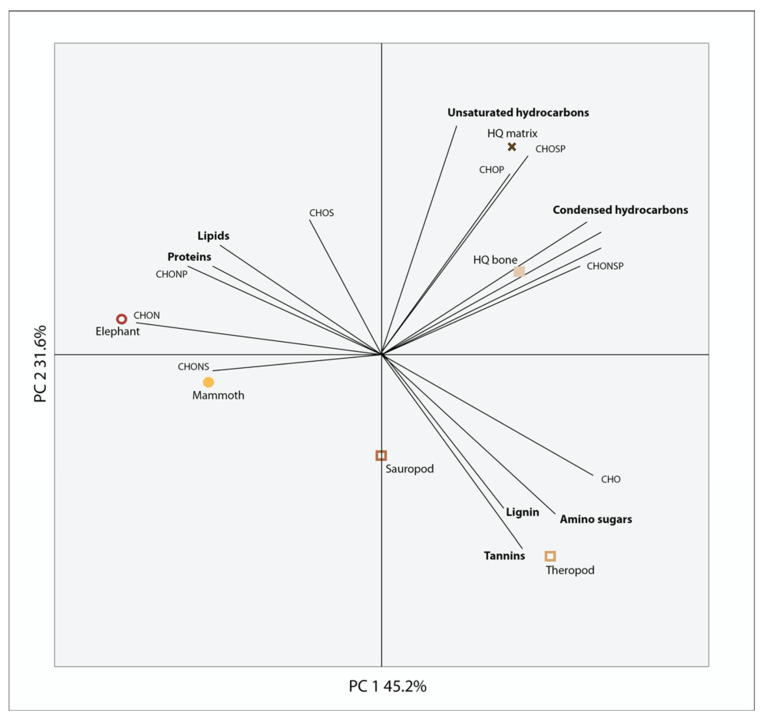
Principal component analysis of FT-ICR MS data. Examination of fossil bones by grinding them up and not demineralizing them reveals additional compounds in greater abundance than proteins and lipids. Proteins and lipids are in the greatest abundance in the elephant and mammoth bones, whereas the Morrison and Hell Creek bones have higher abundances of lignin, tannins and amino sugars and the HQ bones have a higher abundance of unsaturated and condensed hydrocarbons. Amino acids were found in the dinosaur bones and matrix but may be masked by the environmental compounds that are more abundant. Tannins could be any oxygenated compound and lignin could be phenols of plant or microbial origin.

**Table 1 biology-11-01304-t001:** Specimen information and analyses done.

Specimen Number	Specimen	Locality	Time Period	Analyses
GR1063	Coprolite	Hayden Quarry (H2)	Triassic	TOF-SIMS
GR1064	Phytosaur tooth	Hayden Quarry (H2)	Triassic	TOF-SIMS
GR1065	*Tawa hallae* femur	Hayden Quarry (H2)	Triassic	TOF-SIMS
GR1066	Rib fragment	Hayden Quarry (H4)	Triassic	TOF-SIMS, GC-MS, FT-ICR MS
GR1067	Rib fragment	Hayden Quarry (H2)	Triassic	GC-MS, FT-ICR MS
	Matrix sample	Hayden Quarry (H4)	Triassic	TOF-SIMS, GC-MS
LACM 154089	Sauropod rib	Morrison	Jurassic	TOF-SIMS, FT-ICR MS
LACM 23844	Theropod rib	Hell Creek	Cretaceous	TOF-SIMS, FT-ICR MS
	Mammoth rib			FT-ICR MS
TMMM-12613	Alligator rib		Modern	TOF-SIMS
MS-E01	Elephant rib		Modern	TOF-SIMS, FTIR-MS
MS-E01	Elephant rib (experimentally matured)		Modern	TOF-SIMS

## Data Availability

Not applicable.

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
