# Peer review of "Using Macro- and Microscale Preservation in Vertebrate Fossils as Predictors for Molecular Preservation in Fluvial Environments"

_biology, 2022, doi:10.3390/biology11091304_

Round 1

Reviewer 1 Report

Overall, I think this study is important and should be published after rewording the conclusions so they do not overstate the results and clarifying details have been added to almost all sections of the manuscript. Stereotypically, negative results don’t get published but I agree with the authors that we are not going to learn more about the taphonomic controls on preservation until we find and publish on specimens that do not yield biomolecules.

Concerning the conclusions: It is important that the conclusions are completely supported by the results, especially as they are publishing negative results. If you say there is nothing here and are wrong it could have a huge negative impact on the samples future researchers choose to study. However, as written, the authors state that there is no preservation of original biomolecules in their samples, but their methods do not directly address this question. They have not conducted any assays that would identify the amino acids further than amino acids. I have no problem with the assays they used but if this is all you are going to do, you cannot say with certainty that the amino acids are not from an animal or endogenous. I agree it would be questionable for these things to be true in your specific case, but you have not directly tested it so you cannot be definitive in your conclusion. Same goes for the lipids. There is evidence (cited below) that there is lipid exchange between sediment and bone and vice versa. So the signals matching is suspicious but alone does not mean they are not endogenous.

As I said there are many sections of the manuscript that leave the reader asking questions or key details are not included. These details will also be useful to researchers who want to use this study in future surveys of molecular recovery and taphonomic parameters (another key to solving the mystery of molecular preservation). I have indicated these area below in my point by point, as well as other comments and areas of overstated conclusions.

Line 20-21: When you say terrestrial vertebrates do you mean terrestrial fossil or fossil vertebrate specifically?

Line 32-33: I don't understand how testing for molecular preservation is an example of good preservation, especially if the preservation is variable among tested samples. Please rewrite this sentence. I understand and agree with your motivations for the study, selection of site, and specimens, but I am confused by this sentence.

Line 38-40: Just using these mass spec methods isn't enough to put a nail in the coffin of micro vs macro preservation. I suggest a less finite wording than "determined". I have several more comments on this throughout the manuscript and feel strongly that the word should be changed at these locations because the origin of the amino acids has not been directly tested.

Introduction:

-I feel it is relevant to cite Ullmann et al. 2019 since they examined a float fossil for molecular preservation and float bones are generally considered not to have high quality macro preservation as it supports your hypothesis that high quality macro preservation is not a guarantee for molecular recovery. Also, this is the closest anybody has gotten to examining this question which highlights the importance of your study.

Line 63: You might consider adding studies such as Lindgren et al. 2018 and Surmik et al.  2017 that do mass spec on specimens which are from less studied depositional environments (in regards to molecular preservation) - phosphatic chalk and limestone respectively.

Line 74-77: Yes but it might be worth mentioning Lindgren et al. 2018 and Surmik et al. 2016 got molecular recovery from marine depositional environments.

Line 82-84: Please clarify if this is a description of each quarry (as in they all have the same pattern) or if it is an overall description of the variation between the 4 quarries.

Line 85: Please clarify what you mean by "washed in from land". Fluvial strata were deposited in terrestrial settings. Are the quarries interpreted as swamp or lake deposits? I see you cite Irmis et al. 2007 for more details, but it's a bit unclear, and a little more clarity here will prevent most readers from having to look up basic details and will be appreciated.

Materials and Methods:

-Please include your thin section methodology and each samples histological condition (histological index). These should be reported as you talk about micro scale preservation in the introduction and have slides of your samples.

-Why was GC-MS not done on an extant specimen?

Line 117: Do you have any taxonomic identity on this rib? Dinosauria indet. at least?

Line 118-119: It would be nice to specifically state the condition for each sample. Which were float and which were excavated? Which were from mudstone and which were from sandstone? This information could be added to the text or a table (may to table 1?).

Line 122-124: Which elements from each of these taxa? And were these extant specimens analyzed via thin section and whole bone sample?

Line 141-142: Please clarify if you followed Colleary et al. 2021 to mature your samples or if you reanalyzed the same specimens. If you used the same specimens, are there numbers that you can refer to those specimens by so they are cross-referenced?

Figure 1:

-Which are thin sections and which are bone fragments?

-Please label parts A and B and describe B in the figure caption

Line 188: What is H2? I don't see it in your specimen list.

Line 196-197: If this step is important enough to tell us about, please include its purpose.

Line 208-209: Please clarify if you washed the pestle and then wrapped it before combustion or if you washed the aluminum foil and it remained on the pestle through grinding?

Line 213-214: Did you not have enough sample to run these two rib samples separately? Does the taphonomy at the site support that they would have similar preservation? Your description above sounded variable. How do you account for the possibility that one rib might be a better or worse microcosm for molecular preservation?

Line 219-223: Is it possible to list what water soluble and semi polar organics are expected out of bone?

Line 224: Parenthesis typo

Line 240-242: Is this the table in the Supp? If so cite it. If not, please include it.

Line 243: Are these perfect matches or probability alignments? Is this where the directions for lignin and tannins come from in Fig. 4? Should be stated.

Line 248: From a geologic perspective, using the phrase "single depositional setting" to mean any fluvial setting is extremely generalized. There are numerous microenvironmental settings in a fluvial system and each has its own taphonomic mode sensu Behrensmeyer et al. 1992.

Line 265-266: Why is this sentence bolded? Is this supposed to be part of the figure caption?

Figure 2:

-Do you have an image of the bone or could you describe if those circular areas are vessel canals (or any structure of the bone vs. random area of mineral precipitation)?

-Was this sample a fragment or a thin section?

Line 280-281: In looking for biomolecules did you find lipids or to look for biomolecules did you search for lipids? Grammatically something is off in this sentence.

Line 282: This is something you probably want to discuss below but it has been shown that fossil lipids can be leached by ground water out of fossils into sediment (e.g., Liebenau et al. 2015 in Organic Geochemistry) and from sediments into fossil bone (Saitta et al. 2019 in eLife). So just because they are similar does not by itself conclusively indicate the origin of the lipids.

 Line 286: Your specimen list in Table 1 does not include mammoth bones. Please add all samples used to this list for clarity.

 Line 286-287: Were your modern samples degreased before analysis (i.e., treatment in 10% shout)?

 Figure 4:

- HQ bone and matrix are in the same quadrant but not that much further apart than the elephant and mammoth samples, so shouldn’t they be discuss the same way as the mammoth and the elephant? Maybe I missed something here.

 -Lignin and tannins data does not come from modern bone (which is where I assume the rest of the data to assign directions of the biolomolecule types came from). It should be stated where these data came from (what samples, etc.) and all this information should be included in your materials section.

 Line 320-321: Agree. But by saying that you analyzed samples from one depositional environment above you are decreasing support for this argument.

 Line 321-323: Not sure you need to say this again in the discussion but I like the details here and this sentence could be combined with a similar description above or replace the earlier sentence.

Line 328-331: But you haven't looked for any direct evidence that amino acids are animal in origin. You are assuming because you didn't find lipid signatures that differed between the fossils and the sediments that they aren't animal proteins. But that isn't a direct test. It is my opinion that you should be less certain in your conclusions without direct evidence. Language like "our data suggests these my not be amino acids from animals because..." would be more appropriate. This is also important because of the mobility of lipids between sediment and fossils.

Line 332-347: I don't understand why this paragraph is in the discussion. This should be in the methods or introduction to justify your technique - if you feel you need to (other people do this too). They are different techniques with different advantages (as you state very well). They are not interchangeable nor mutually exclusive. So I'm not sure this paragraph is required but keep it if you want.

Line 357-359: Wouldn't this be supportive that what you found isn't general contamination because it is specific to bones and similar for samples from the same place?

Line 359-362: Could you explain this more? Why and what significance is this to the fossils that plot near it?

Line 379: In addition to what?

Line 390-392: Please discuss what the ubiquitous iron and isolated strontium means in terms of preservation expectations, diagensiss, and/or taphonomy.

Line 392-394: I don't get why you would expect a bone from a coprolite to be that different from a fossil bone - assuming anything could survive digestion and fossilization.

Line 394-396: But the environments that form mudstones are not similar to those that form sandstones so you are analyzing fossils from different environments and thus would not necessarily expect the same taphonomic processes occurring.

Line 399-400: This should instead be something like "no direct evidence for..." or "our results question the presence of original biomolecules" because you did find evidence of biomolecules, but have not directly tested their origin. Therefore, you have no evidence from lack of testing, not falsified hypotheses, and your conclusions should be stated as such. Don't mean to sound harsh, but it is important to not overstate what you can conclude from your results.

Line 400-402: I agree but it is suggestive of it and more testing should be done to identify these amino acids and their origin before definitively concluding that they are contamination. I'm not saying they aren't contamination, just that you haven't definitively shown this.

Line 402-406: To my knowledge, it has not yet been tested if the presence of lipids correlates with the presence of proteins in fossils (similar to how Poinar and Paabo 2001 suggest for DNA). If it has, please cite that study. I agree that not finding lipids makes the origin of the amino acids questionable but without further testing, I don't agree with stating it as fact and from one sample your results should not be generalized as fact to definitively have or not have molecular preservation. One sample is not enough to make broad reaching conclusions. It can be “suggestive” or “indicate that ________ is possible”.

Line 408-413: This is an interesting and important result even if it is mostly a negative result.

Supplemental Material: The tables in the Supplemental Material need more explanation. There are not even table captions.

Reviewer 2 Report

The paper presents the results of a series of mass spectrometry techniques carried out on the exceptionally preserved material from Hayden Quarry. The authors thereby develop and establish procedures to apply the combination of TOF-SIMS, GC-MS, and FTIRC-MS as a standard testing scheme for taphonomic preservation of biomolecules.

The methodology of the approach is exciting. It is therefore unfortunate, that the authors were unable to demonstrate the endogeneous origin of the biomolecules in the material submitted to analysis.

There are no issues concerning the contents of the publication. However, in a few instances, there are problems with the formatting of the text starting with the paragraph above Fig. 2 on page 7, which has been apparently mistaken with the figure legend under Fig. 2, and vice versa. Font size is not correct.

The legend for Fig. 4 on page 9 uses also a wrong font size, the figure itself has a strange black margin and causes a page break in its present size.

Font sizes of the section 'Attribution' and in the section 'Acknowledgments', both on page 11, are wrong.

Finally, two of the references (both by the first author Tfaily) are not indented properly.

Reviewer 3 Report

My comments can be found in the attached file
